# Mapping the Chemistry of Hair Strands by Mass Spectrometry Imaging—A Review

**DOI:** 10.3390/molecules26247522

**Published:** 2021-12-11

**Authors:** Mai H. Philipsen, Emma R. Haxen, Auraya Manaprasertsak, Per Malmberg, Emma U. Hammarlund

**Affiliations:** 1Lund University Cancer Center, Department of Laboratory Medicine, Lund University, 221 00 Lund, Sweden; hmai@chalmers.se (M.H.P.); emma.haxen@med.lu.se (E.R.H.); auraya.manaprasertsak@med.lu.se (A.M.); 2Lund Stem Cell Center, Department of Laboratory Medicine, Lund University, 221 84 Lund, Sweden; 3Department of Chemistry and Chemical Engineering, Chalmers University of Technology, 412 96 Gothenburg, Sweden; malmper@chalmers.se

**Keywords:** mass spectrometry imaging, hair analysis, sample preparation

## Abstract

Hair can record chemical information reflecting our living conditions, and, therefore, strands of hair have become a potent analytical target within the biological and forensic sciences. While early efforts focused on analyzing complete hair strands in bulk, high spatial resolution mass spectrometry imaging (MSI) has recently come to the forefront of chemical hair-strand analysis. MSI techniques offer a localized analysis, requiring fewer de-contamination procedures per default and making it possible to map the distribution of analytes on and within individual hair strands. Applying the techniques to hair samples has proven particularly useful in investigations quantifying the exposure to, and uptake of, toxins or drugs. Overall, MSI, combined with optimized sample preparation protocols, has improved precision and accuracy for identifying several elemental and molecular species in single strands of hair. Here, we review different sample preparation protocols and use cases with a view to make the methodology more accessible to researchers outside of the field of forensic science. We conclude that—although some challenges remain, including contamination issues and matrix effects—MSI offers unique opportunities for obtaining highly resolved spatial information of several compounds simultaneously across hair surfaces.

## 1. Introduction

Strands of human hair preserve chemical components from our environment and reflect our living conditions and lifestyle. For example, environmental mercury pollution can be detected in the hair of humans that have a diet rich in fish [1]. Strontium in our drinking water reflects the distance between the freshwater source and the ocean and can, after becoming preserved in hair, assist forensic investigators in linking unidentified victims to the geographic area in which they lived [2,3]. Additionally, drug use is recorded in hair and can be detected with high temporal precision since hair grows at a known rate [4,5]. Hair, as a historic record of the chemistry of our living conditions, offers some advantages over samples of blood and urine. While analyses of blood and urine provide short-term information (2–4 days) related to, for example, drug intake [6,7], hair-strand analysis offers a longer window for the detection of drug use, ranging from days to years [8]. In addition, hair can be sampled non-invasively. However, analysis of hair chemistry also has its challenges, such as those associated with the risks of contamination.

Techniques for the chemical analysis of hair have traditionally involved bulk analyses of entire hair strands using chromatography and inductively coupled plasma mass spectrometry (ICP-MS) [9,10,11]. Such bulk analyses are complicated by the fact that hair products and dust can introduce exogenous chemical signals and thus interfere with the overall interpretation [10,12]. Therefore, preparation procedures, such as different washing steps, have been employed to remove superficial contaminants. However, these preparations are time consuming and may be associated with the risk of losing endogenous compounds of interest [12,13]. The need for large amounts of sample and time-consuming preparation procedures, combined with the fact that these methods do not easily offer temporal resolution of data, have led to the development of improved methodologies. 

Since the early focus on bulk analyses, the field has turned to analytical techniques that provide high spatial resolution analysis of elemental and molecular species in strands of hair. Spatial mapping of chemical compounds in hair can be achieved through electron beam methods, microscopy, and nuclear reaction analysis, but these techniques provide information about metallic elements, not larger biomolecules [14,15]. Most recently, MSI methods, including secondary ion mass spectrometry (SIMS) and matrix-assisted laser desorption/ionization mass spectrometry (MALDI-MS), have proven to be viable alternatives for mapping the chemical and elemental landscape of individual hair strands [16,17]. These methods enable direct identification and imaging of several molecules and their metabolites in intact and sectioned hair. However, using MSI techniques for hair-strand analyses requires the development of specific protocols related to sample preparation. 

When analyzing hair chemistry, the sample preparation procedures involved depend on which part of the hair is intended for analysis. Hair is made up of the cuticle, cortex, and—when mature and thick enough—the medulla (Figure 1) [18]. Human hair fibers are covered by multiple layers of cuticle cells that protect the inner structure from damage caused by external factors, including environmental agents, cosmetic treatments, and industrial processes [19]. This inner structure is mainly made up of the cortex, which consists of several intermediate keratin filaments, cross-linked with keratin-associated proteins via extensive disulfide bonds [20,21]. The hair cortex provides hair strength, moisture, and color, whereas texture is related to the properties of the hair surface [22]. The innermost structure is the medulla which is usually only present in coarse hair, such as thick hair and beard hair [23]. As will be outlined in the following, some areas of study focus on the chemical composition of the cuticle, whereas others are more concerned with hair-core analysis (i.e., the composition of the cortex and medulla). Consequently, different sample preparation procedures have been developed for hair-cuticle and -core analyses via MSI.

Here, we review the steps involved in MSI analysis of cuticle and core, focusing on sample preparation procedures. We present how different washing and cutting methods have been used in hair-sample preparation to minimize chemical losses and to improve the analytical potential that these techniques have to offer. The review is intended to function as an introduction to the current state of the art and use cases, making the methodology more accessible to researchers within fields that do not have a tradition for the analysis of hair chemistry.

## 2. Techniques over Time—From Bulk Analysis to High Resolution Mapping 

Analyses of hair chemistry have been brought forward by the forensic and clinical toxicology sciences [24]. During the 1960s and 1970s, the concentration of heavy metals in hair strands was quantified by atomic absorption spectroscopy [25,26,27]. Subsequently, the sensitivity and speed of quantification of elemental concentrations were improved through inductively coupled plasma (ICP) emission microscopy. Especially ICP coupled with mass spectrometry (ICP-MS) had a profound impact on trace-element analyses in scientific disciplines, from geology to forensic medicine [28,29,30]. In 1979, it was demonstrated that morphine could be detected in hair from heroin users using radioimmunoassay [31]. Since the mid-1980s, chromatography of hair has commonly been used in the analysis of drug use, especially when coupled with mass spectrometry and tandem mass spectrometry [32,33,34,35,36,37]. Such bulk methods enabled the construction of a temporal record when several strands of hair were aligned in the direction of growth and then cut into segments of 1–2 cm. The segments were analyzed separately and the chemical information was puzzled back into the order of growth, allowing weeks-to-months-long records of drug use to be constructed [38]. These bulk analysis techniques have led to valuable insights but require a significant amount of sample as well as time for digestion and extractions. It is particularly time-consuming to determine the spatial localization of compounds in hair and, therefore, the temporal distribution of exposure to the compounds.

Imaging techniques have significantly assisted the field to acquire high-resolution maps of the chemical landscape of hair. Several imaging approaches enable high-resolution mapping of hair, such as nuclear microscopy, Raman scattering microscopy, and synchrotron radiation micro-X-ray fluorescence [14,39,40]. Although offering spatial mapping of several compounds, these methods are limited to detecting specific elements and, particularly, large molecules in hair. More recently, MSI has been applied to obtain the spatial distribution of several species on and in hair, ranging from small elements to molecules as large as 4000 Da [41,42,43], with a low limit of detection and high spatial resolution down to 50 nm [44]. This is achieved by applying different so-called mass analyzers, enabling MSI to resolve different masses (see Table 1 for a general comparison of MSI mass analyzers). With mass analyzers, such as orbitrap or Fourier transform ion cyclotron resonance (FTICR), the mass resolution can range from 6000 with time-of-flight (TOF) instruments to more than 100,000 with ultra-high resolution [45,46]. Mass resolution is the minimum separation between two mass spectral peaks defined as m_1_/(m_2_−m_1_). The high surface sensitivity and lateral resolution, combined with low detection limits and the capability of detecting both organic and elemental species, make MSI techniques uniquely suited for spatial mapping of analytes in hair. 

Due to their high spatial resolution, MSI techniques have emerged as a great tool to address issues in forensic investigations and investigations into the effects of cosmetic treatments [17,47]. For example, the techniques have been used in reconstructing temporal records of drug abuse, based on imaging of drugs in longitudinal hair sections [48,49,50], and in determining age-related changes in biomolecular compounds in the hair cortex [42]. Importantly, MSI of hair makes it possible to distinguish drugs incorporated via the bloodstream from drugs introduced from exogenous sources [17,47]. Analysis of intact hair strands is possible without any further chemical processing, and because damage to the samples during MSI experiments is usually minimal, downstream analyses with other approaches are possible. Additionally, MSI is made attractive by requiring less sample material and less intensive sample preparation procedures without the requirement for isotopic labeling of compounds [51]. However, as will be described in the following, the techniques also have their limitations and pitfalls that need to be considered when planning MSI analyses.

**Table 1 molecules-26-07522-t001:** General comparison of several common mass analyzers used in MSI. Data obtained from [52,53].

	TOF Reflectron	Magnetic Sector	Orbitrap	FTICR
Primary ion	Pulsed	Continuous	Pulsed	Pulsed
Upper mass limit	10,000	20,000	50,000	30,000
Mass resolution	15,000	<100,000	>100,000	1,000,000
Mass accuracy	<5 ppm	<3 ppm	<5 ppm	<1 ppm
MS/MS	MS	MS^2^	No	MS^n^
Advantages	Good mass accuracy Fast scan speed	High mass accuracy and resolution	High mass accuracy and resolution	High mass accuracy and resolution
Drawbacks	Medium mass resolution	Expensive	Expensive	Low scan speed Expensive

## 3. Analysis of Hair by Mass Spectrometry Imaging

Traditionally, MSI techniques have been most widely used in the materials sciences. However, since MSI can be used to analyze and visualize any chemical species that can be desorbed and ionized from a sample surface, the methods have since found application within the biomedical and forensic sciences as well. The three major ionization techniques commonly used in MSI are MALDI, SIMS, and desorption electrospray ionization (DESI) (Figure 2). 

SIMS was the first ionization technique to be developed and is, therefore, the oldest MSI method [54,55]. In this approach, a high-energy primary ion beam is applied to sputter the sample surface, resulting in the generation of ionized species that are then separated based on their mass-to-charge ratio (*m/z*) using a TOF or magnetic sector mass analyzer. Traditional SIMS instruments were operated with high doses of a monoatomic ion beam (e.g., Ar^+^, Ga^+^, and Bi^+^) which damaged the sample surface and produced small fragment species [54,56]. Thus, the detection of intact biomolecules was limited with SIMS-based imaging. However, subsequent evolution of primary ion beams—from monoatomic ion beams to cluster ion beams (Bi_3_^+^, Au_3_^+^, C_60_^+^, and so on) and, later, gas cluster ion beams (Ar_4000_^+^ or (CO_2_)_6000_^+^)—has since enabled SIMS analysis of intact molecular ions [57,58,59]. In cluster ion beams, the kinetic energy is divided between several atoms, resulting in lower energy of individual particles. Consequently, the degree of molecular fragmentation and subsurface damage is reduced, improving the mass range of detection for heavier species up to 2500 Da [60]. However, despite these advantages, SIMS remains of limited utility in the analysis of larger biomolecules, such as proteins and peptides.

While SIMS has gained attraction in recent years, MALDI is, at present, the most commonly used technique for biological applications. For this method, a matrix compound, typically an organic acid, is deposited on the sample surface prior to analysis in order to facilitate desorption and ionization. After a laser beam strikes the matrix-coated sample surface, the matrix molecules absorb the laser energy and convert it into heat energy. A fraction of the matrix molecules from the top layer of the sample surface is then vaporized along with analytes, which are ionized via ion or charge transfer processes [61]. In some cases, ionization efficiency can be improved by so-called derivatization, where the sample chemistry is altered in order to change the properties of the analyte. For example, Beasley et al. demonstrated that in situ derivatization improved ionization efficiency enough to enable the imaging of cannabinoids in single hair samples [62]. The popularity of the MALDI technique reflects its ability to probe a variety of molecules, including lipids, proteins, peptides, nucleotides, and saccharides [63,64,65,66]. However, it is to be noted that the application of matrix to the sample surface can interfere with the detection of low-molecular-mass species (<600 Da) [67,68]. Moreover, reproducibility and spatial resolution is limited by the matrix crystal size, raster step size, and laser beam diameter.

Like MALDI, DESI is an ionization technique suitable for the analysis of biological samples. In DESI, the sample surface is bombarded with electrically charged solvent droplets to desorb analytes of interest, which are ionized using electrospray. The ionized molecules then travel into an inlet capillary towards mass analyzers for analysis. The main advantage of this technique is that desorption and ionization can take place under ambient conditions—in contrast to SIMS and MALDI, which operate under vacuum—without sample preparation or matrix application. Consequently, it is a comparably fast technique that makes it possible to preserve the physical and chemical properties of the sample. Today, DESI has great potential to aid forensic investigations in the screening for and identification of drugs [69]. However, the technique has not been extensively used for MSI analysis of hair, probably because of its relatively low spatial resolution. Recent advances have led to improved resolutions of between 20 and 100 µm [70,71], but the average diameter of adult human hair varies between 45 and 110 µm [18]. As the technique evolves towards improved resolution [70], it may find application in future hair analyses.

When planning MSI analysis of hair samples—or any other sample—it is important to choose an appropriate ionization process. As described above, different techniques are suitable for different analytes, with MALDI covering a wider mass range (1–500 kDa) than SIMS. Another consideration is resolution requirements. As SIMS utilizes an ion beam as opposed to a laser, this method has a spatial resolution as high as 100 nm [72], whereas the highest resolution offered by MALDI is 5 µm [73]. The higher resolution makes SIMS the more suitable technique for imaging the detailed distribution of compounds within the hair structure, but it also increases the risk of obtaining unfocused images that might lead to misinterpretation [17]. When imaging large molecules >2000 Da, MALDI is the best method for hair analysis at present, but the addition of the analyte-specific matrix alters the sample composition and may lead to the relocation of compounds [74]. Because MALDI and SIMS have different strengths and weaknesses, applying both methods makes it possible to obtain complementary information [75]. However, whether the choice falls on one or both of these techniques for a given analysis, it is important to be aware of their limitations and challenges.

One challenge in MSI analysis of hair strands is that hair samples are non-conductive, and, therefore, it is important to mount the hair on conductive tape along its entire length for MSI analysis in order to minimize charge build-up. However, studies have demonstrated that hair can become contaminated by the tape substrate leading to misinterpretation of data. For example, the identification of dimethicone, a silicone polymer used as a softening agent in shampoos, is challenging because the surface of most materials, as well as the double-sided conductive tapes used to mount samples, contains silicone [16]. Furthermore, poor contact with the conductive tape leads to poor signals and bad image quality. To address this, conductive tape with a smoother surface, such as silver and copper tapes, can be used to get better overall contact [16,76]. However, this issue is only relevant when analyzing intact or longitudinally sectioned hair strands, illustrating that different types of analyses are associated with different pitfalls.

Another challenge when conducting MSI analyses is that the techniques only provide qualitative or semiquantitative data. This is because the sample matrix influences the ionization probability of analytes, resulting in enhancement or suppression of the signal intensity [77]. These matrix effects are more severe for organic compounds where protonation and charge transfer processes occur [78]. This can be mitigated to some extent by chemical modification of the sample [79] or by coating the sample surface with a layer of matrix prior to MSI experiments. For example, in SIMS, samples can be coated with a matrix to increase the signal to noise ratio of analytes [80]. In MALDI, the concept of matrix effects is exploited, and a matrix compound is deposited on the sample surface to mediate desorption and ionization of a given analyte [55]. For example, Wang et al. used umbelliferone to improve the detection limit of methamphetamine in hairs down to nanogram per milligram using MALDI-FTICR [81]. The most appropriate type of matrix solution and method of deposition should be evaluated for each study [50]. In general, the limitations and pitfalls of MSI can be addressed through the application of appropriate sample preparation procedures, and choosing the correct procedure is critical for the accuracy of results obtained through MSI. 

## 4. Sample Preparation Procedures: Considerations, Pitfalls, and Use Cases

Single-hair analysis has been developed for different purposes, from hair cuticle analysis to hair core analysis using both cross- and longitudinal sections. A work chart summarizing the different sample preparation procedures for MSI techniques is presented in Figure 3. In the following, we will discuss the different sample preparation procedures and describe relevant use cases in hair core and cuticle analysis.

### 4.1. Hair Core Analysis

The inner structure of hair, consisting of the cortex and medulla, is formed by cells and molecules from the local tissue but also from the bloodstream [82]. Since our intake of food and drugs are reflected in the bloodstream, the chemistry of hair can capture aspects of our lifestyle. For example, MALDI has been applied to detect and image nicotine in longitudinal sections of hair from heavy smokers [83]. However, hair chemistry does not only reflect direct endogenous uptake from the bloodstream; it can also reflect indirect uptake via sweat and sebum or contamination from external sources such as vapors or powders. Consequently, when analyzing hair chemistry, it is important to distinguish between endogenous and exogenous compounds. This distinction is notoriously challenging, both in bulk and MSI analyses, and several approaches have been attempted. For bulk analyses, these include different washing protocols, and for MSI techniques, different methods of sectioning have been proposed as well. In the following, we describe how the issue of contamination can be dealt with and specify procedures for both washing and sectioning of hair samples prior to MSI analysis of the hair core. 

#### 4.1.1. Washing Procedures 

Mitigating the risk of exogenous contamination of the hair core is critical. Indeed, in the case of forensic analyses, contamination may have severe implications for people’s lives. The risk of contamination is typically addressed by washing samples prior to analysis, but this step is more critical in bulk and cuticle analyses, as these do not enable the distinction between the outer and inner structure of the hair. However, even when analyzing sectioned hairs using MSI, some risk of contamination remains since the hair cortex may get in contact with exogenous compounds during sectioning. Thus, it is still generally recommended to wash samples prior to sectioning and MSI analysis, and care must be taken to choose an appropriate washing method that will remove exogenous contaminants while retaining endogenous compounds. Several decontamination procedures have been proposed, as reviewed by Kempson and Skinner in 2012 [84]. Using TOF-SIMS, these authors also investigated the effects on elemental components of two different washing procedures, namely the Internal Atomic Energy Agency (IAEA) recommended procedure and a detergent-based procedure involving a 2% Triton X-100 solution [84]. It was found that most exogenous elements, including Na, K, Ca, and Mg, were removed through both washing procedures, whereas the signal of exogeneous Fe was unaffected. Simultaneously, it was observed that the signals of these elements also decreased in the internal structure of hairs. This signal reduction might reflect either the loss of endogenous elements or the removal of contaminants that had penetrated into the hairs from exogenous sources during washing. As highlighted by this study, a fully appropriate standardized procedure does not exist, and it is therefore up to the individual lab to develop appropriate washing protocols and investigate their capacity for removing contamination [85].

Washing procedures are discussed by, for example, the Society of Hair Testing which generally recommends washing samples with organic solvents, followed by aqueous rinses [86]. The purpose of washing is not only to remove contaminants, but also to clean the sample of surface materials, such as hair products, sweat, and sebum, which may influence the analysis [85]. Recently, in 2016, Cuypers et al. applied several washing steps in an attempt to distinguish between exogenous and endogenous cocaine in hair samples using MALDI-MS/MS and metal-assisted SIMS [87]. Using methanol, water, dichloromethane, hexane, and acetonitrile, these authors washed hair samples that were contaminated by exogenous cocaine, as well as hair samples that were enriched in endogenous cocaine. It was found that, in this case, rinsing with water and methanol was apparently more efficient in reducing exogenous contamination of cocaine than rinsing with other solvents, as seen in Figure 4. In contrast, Erne et al. reported that washing hair samples with dichloromethane and methanol failed to completely remove exogenous contamination of the drug zolpidem (used for insomnia treatment), resulting in higher signals of zolpidem in contaminated intact hair compared to the negative control [47]. To solve this issue, they proposed using a phosphate buffer and several short methanol washes instead to completely remove zolpidem from intact hairs. The cases described here illustrate the difficulty in developing standardized washing protocols, as different compounds call for different washing procedures. 

Another potential issue that should be considered when developing washing procedures relates to the migration of external contaminants into the hair core. In theory, sectioning hair samples prior to MSI analysis enables the distinction between exogenous and endogenous compounds, located in the superficial and inner parts of the hair, respectively. However, recent studies have suggested that washing might cause the migration of contaminants into the hair cortex and medulla [47,87,88]. This is because the moisture induces swelling of the cuticle, which allows chemicals to diffuse into the interior parts of the hair [88]. The migration of, for example, external cocaine into the hair core makes it difficult to discriminate between hair from actual cocaine users and hair containing external contamination [87]. Such external contamination can come from powders or vapors that have diffused into the hair, but it might also come from indirect uptake through sweat and sebum, which can complicate the reconstruction of a temporal record of consumption. Studies have demonstrated that even a single instance of drug use can lead to drugs becoming incorporated into hair strands through sweat and sebum [5,88,89]. Such contamination obscures the temporal record of drug intake and can make it difficult to distinguish between single cases of drug use and chronic consumption [89]. The fact that external contaminants can become incorporated into the inner structures of hair poses a potentially very severe problem for the concept of hair analyses.

To address these concerns, the decontamination protocol should not only be carefully chosen depending on the specific analytes but the effects should also be evaluated in individual experiments to avoid misinterpretation of results. Recently, Erne et al. examined the effects of different washing procedures on the content of zolpidem in hair strands to determine whether decontamination procedures might help to distinguish between washed-in contamination and endogenous compounds [47]. These authors demonstrated that a developed in-house washing protocol was able to remove zolpidem that had migrated into the hair strand, while not erasing the signal of endogenous zolpidem. Similarly, subsequent studies also suggest that future sample preparation protocols might be able to address the issue of incorporated contaminants, making it possible to distinguish between endogenous and exogenous compounds within the hair core [17]. However, such protocols are analyte-specific, and further studies are needed to develop decontamination procedures that can remove exogenous contamination without causing the incorporation of contaminants. This issue further highlights the need for carefully examining the effects of decontamination procedures when developing sample preparation protocols. 

#### 4.1.2. Cross-Sectioning of Hair Strands

Cross-sections of hair strands have been used in several elemental and molecular analyses [90,91,92,93]. In the past, morphological characteristics of cross-sectioned hairs were used by forensic investigators to evaluate the differences between hairs from different parts of the body and from different individuals. Now, MSI offers mapping of the spatial distribution of chemical compounds within the hair. In general, there are two approaches when working with cross-sectioned hair: one in which the hair sample is fixed using an embedding medium prior to cutting, and one in which the hair sample is cut without the use of chemical embedding. Both approaches have their advantages and limitations, as will be described in the following.

To achieve consistent and representative cross-sections of hair strands, samples can be pre-embedded using specific materials in order to minimize cuticle swelling and obtain a true cross-section. The embedding step is then followed by sectioning using a microtome, which lowers the risk of the cutting blade distorting the hair sections and enables more precise and repeatable analyses. Embedding media usually used for cross-sectioning of hair include celloidin, paraffin, or resins [14,16,42,87,94]. Hairs embedded in paraffin alone can bend at the cutting edge, leading to the cross-section becoming elliptical rather than circular [95]. In contrast, the morphology of hair cross-sections is preserved better using celloidin embedding [96]. Hair cross-sections with true morphology can also be obtained through resin embedding, but the hair sections might separate from the resin during sectioning [97], and resin has been observed to infiltrate into the medulla [98]. The effects of other embedding media, including carboxymethylcellulose, gelatin, and trehalose, have also been investigated, specifically for methoxyphenamine detection in hair cross-sections, as shown in Figure 5 [50]. In this case, the use of gelatin yielded the best results with less contamination and sample distortion, as well as low background noise. Other studies have also found that the use of gelatin maintains sample morphology while eliminating background interferences [99,100]. Recently, Flinders et al. (2017) used gelatin as an embedding material prior to MSI analysis, detecting cocaine and its metabolite with low background noise in cross-sections of hairs from drug users [76]. However, although microtome sectioning of embedded samples preserves the morphology of hair sections and is widely used, these cutting methods are also associated with their own specific limitations. The preparation procedure for embedding and microtome sectioning is time-consuming, and embedding chemicals may lead to contamination during cutting [101]. 

Due to the risk of contamination, sectioning hair without chemical embedding is preferred. One simple way to prepare cross-sections of hair without an embedding medium is to directly cut hair mounted on tape using a razor held at an angle [16]. It is, however, very difficult to cut loose hairs without damage. Another concern is that the hair morphology may be altered by the razor blade, and, to counter this risk, Gillen et al. reported using aluminum micro vises for cross-sectioning of hair samples [16]. A bundle of hairs was placed between the two sides of the vise which was then inserted into the sample holder block. The vise was adjusted in order to position the hairs perpendicularly to the sample holder surface, and hairs sticking out of the vise were cut by a razor blade to expose a fresh cross-section for MSI analysis. The micro vise has an optimal design that produces a smoother cross-section surface and minimizes damage to the hair.

#### 4.1.3. Longitudinal Sectioning of Hair Strands

Compared to cross-sectioning, longitudinal sectioning of hair makes it possible to analyze a broader cortex surface area, providing more information per analysis. This method of sectioning also circumvents the issue of analytical uncertainty related to the fact that the shape of cross-sections along the length of the hair can vary between circular, triangular, irregular, or flattened [102,103,104]. Studies have demonstrated that longitudinal sectioning of hair enables the identification of analyte accumulation patterns which may reveal whether compounds are contaminants or endogenous to the hair medulla [50,105]. Additionally, using longitudinal sectioning makes it possible to section only part of the hair strand and examine both intact and sectioned segments within a single hair to distinguish between exogenous and endogenous compounds [47,106]. Consequently, longitudinal sectioning is typically used in forensic investigations, where it is relevant to determine when, or for how long, a drug has been consumed [8,48,49,50], or in studies examining the effects of contamination [47,87]. Combined imaging of cross- and longitudinal sections makes it possible to investigate the three-dimensional distribution of compounds within the sample [50,87].

Several methods for longitudinal sectioning of hair strands have been published. For example, Shen et al. gently scraped off the hair surface using a scalpel, from the proximal to the distal end, to image ketamine in hair sampled from chronic users by MALDI [49]. Although the sensitivity of the analysis was improved compared to intact hair, the repeatability was low. In another study, Miki et al. also used MALDI on longitudinal sections to demonstrate the incorporation of methamphetamine into human hair [48]. Here, hair samples were affixed lengthwise to, and half-embedded in, a piece of narrow carbon tape attached to an indium tin oxide-coated glass slide. The authors then compared manual sectioning, using a razor, to laser sectioning, demonstrating that the laser beam provided a fast and significantly more accurate cutting than the manual approach. However, this method also led to the loss of methamphetamine, probably due to the volatile properties of the analyte and/or thermal denaturation of the hair, leading the authors to prefer manual sectioning. Even so, a major challenge associated with the manual longitudinal sectioning of hair is low reproducibility. 

The issues with accuracy and repeatability of longitudinal sections cannot be solved using chemical embedding due to sample size and direction control limitations when operating a microtome; embedding the length of a hair sample within a medium makes it near impossible to ensure that the microtome blade splits the hair exactly through its center. However, many recent studies have used freeze sectioning with ice-embedded hairs for longitudinal sections [50,83,103]. For example, Kamata et al. optimized the cutting procedure with ice embedding to examine the distribution of methoxyphenamine in a single hair shaft and root [107]. The hair was cut to a length of approximately 30 mm and half-embedded in conductive adhesive tape on an indium tin oxide-coated glass slide by pressing with an aluminum block wrapped in film. Several drops of distilled water were added onto the hair which was subsequently frozen at −20 °C. The hair sample was then freeze-sectioned using a rotary microtome equipped with a retraction system. The advantages of frozen sectioning include its simplicity and improved repeatability, but the method is also associated with a number of problems, such as freezing artefacts and uneven sample embedding. These issues highlight the need for the development of new methods to facilitate accurate and reproducible longitudinal sectioning of hair samples.

One solution proposed to address the issues mentioned in the above is the development of a new device for the preparation of longitudinal hair sections without embedding. Such a device was first proposed by Kempson et al. and subsequently refined by Flinders et al. [106,108]. It consists of a cutting device and a stainless-steel cutting block containing several grooves with depths of 20–80 µm (Figure 6). Having measured the width of the hair using a digital micrometer, the sample is placed into a groove whose depth corresponds to half of the hair diameter. At one end, the hair is secured by tape, and at the other, it is held by a gloved finger. The cutting device is constructed so that a microtome blade cuts along the length of the hair at an angle of ~20°, and this method ensures greater control and reproducibility than what can be obtained by manual cutting methods. Additionally, this cutting technique is very rapid and does not require an embedding medium to be used. Using such a device for hair sectioning facilitates the examination of drug distribution in longitudinal sections by, for example, metal-assisted SIMS, as shown in Figure 7 [108]. Some difficulties, however, remain to be overcome. For example, the method requires hair samples to have a diameter of at least 30 µm and a length of at least 1 cm. As a result, this technique is not suitable for hairs thinner than 30 µm or for very short hair samples (such as samples from animals with short fur). However, despite these limitations, this approach is deemed the preferred method for obtaining lengthwise sections of hair at present.

### 4.2. Hair Cuticle Analysis 

The hair cuticle layer is the outermost layer of the hair shaft, where all hair treatments are first applied. It consists of overlapping cuticle cells protecting the hair interior [23], each about 0.5 µm thick and 45–60 µm long [18]. The individual cells consist of a thin outer membrane (the epicuticle), the A-layer, the exocuticle, and the endocuticle (Figure 1). Because the hair cuticle is the layer most affected by the components of hair products, such as dye molecules, several hair cuticle analyses focus on the effects of cosmetic treatments. For example, Kojima et al. investigated the penetration and distribution of dyes in the hair cuticle using Nanoscale SIMS, finding that the endocuticle contained more dye molecules than the other cuticle structures [109]. Such studies help to shed light on the mechanisms underlying different cosmetic treatments, providing insights that are relevant to the development of, for example, hair dyeing products. 

Another purpose of hair cuticle analysis is to provide a quick and simple method for the detection of drugs and pharmaceuticals. Several studies have proposed the use of MSI for the detection of drugs in intact hairs for forensic investigations. For example, Porta et al. imaged the distribution of cocaine and its metabolites in single intact hair strands using MALDI, detecting concentrations down to 5 ng/mg (Figure 8) [110]. Additionally, MALDI-MS/MS has been used in the detection of zolpidem in single hair strands [111], and the synthetic opioid painkiller tilidine has been imaged in intact hairs of children by MALDI-MS for a forensic case [89]. More recently, MSI has also been used in the analysis of different pharmaceuticals in hair strands, enabling a measure of medication adherence. For example, infrared MALDI electrospray ionization MSI has been used on intact hairs to simultaneously detect multiple antiretroviral medications administrated for the prevention and treatment of HIV/AIDS [112,113]. However, issues relating to poor resolution and sensitivity pose potential problems for hair cuticle analysis, since endogenous drugs are mainly incorporated into the core of the hair strand through the blood stream. Consequently, while the results of these and other studies have provided valuable contributions to our understanding of the mechanisms underlying drug incorporation in hair, some limitations of MSI for hair cuticle analysis still need to be resolved.

One of the challenges currently facing hair cuticle analysis using MSI techniques is how best to mitigate the effects of surficial contamination. Because hair cuticle analysis alone does not provide insight into the spatial distribution of compounds within the hair, removing exogenous contaminants is especially important for this type of analysis. Suitable decontamination procedures that are able to completely remove exogenous interferences—such as those stemming from sweat, sebum, dirt, and other surface contaminants—are needed in order to improve the signal of specific analytes and reduce background noise. As removing surface contamination is especially important in MSI analysis of the hair cuticle, sample preparation methods usually include washing procedures similar to those for bulk analyses [110,111]. A variety of decontamination procedures have been proposed, including washing with methanol, ethanol, dichloromethane, isopropanol, or buffers [36,37,115,116,117,118,119]. However, many methods might lead to the loss of several elements and molecules, affecting total concentrations [12,115].

Due to this issue, it is necessary to critically examine the effects of different washing procedures for hair cuticle analysis as well. For example, Borella et al. evaluated the washing efficiency of six different cleaning procedures on nine trace elements in the hair [13]. The authors found that EDTA wash removed more trace elements than detergents and organic solvents did, whereas the loss of Cr and Cd was higher using sodium lauryl sulphate and methods recommended by the IAEA. A risk of contamination by rinsing with an acetone/methanol mixture was also reported. Bossers et al. found about 50% loss of alcohol markers in hair samples using a rinsing procedure with dichloromethane followed by methanol [120]. Additionally, washing with hot ethanol has been shown to remove the outer membrane of cuticle cells of human hair [121]. These examples highlight the need for caution when choosing washing procedures to avoid the risk of losing the outer layer of the cuticle membrane or any compounds of interest on the hair surface. As different decontamination procedures are appropriate for different analytes and purposes, it is necessary to carefully evaluate the suitability of a given washing method for each individual study.

## 5. Conclusions

MSI techniques, like MALDI-MS and SIMS, enable high-resolution spatial mapping of elemental and molecular species on and within individual hair strands. The methods complement traditional bulk analyses and provide unique insights into, for example, mechanisms of drug uptake. However, in order to ensure the accurate interpretation of results, appropriate preparation of samples prior to analysis is crucial. Therefore, in this review, we have presented and discussed different sample preparation procedures, focusing on decontamination steps and sample sectioning. 

The decontamination process is critical in hair analysis, especially within the field of forensic science. Ideally, it serves to remove external interferences while preserving endogenous compounds of interest. When the focus of the analysis is chemicals on the hair surface, like in hair cuticle analysis, developing appropriate decontamination protocols is especially important for obtaining reliable results. Several washing procedures have been suggested for removing different surface contaminants, but appropriate washing steps are analyte-specific and should be determined depending on the purpose and type of the analysis. Evaluating the impact of washing steps is especially important since studies have shown that some organic solvents can wash out internal compounds, whereas other procedures promote the migration of contaminants into the hair. 

In contrast to bulk analyses, MSI analysis of sectioned hair can, in theory, enable the distinction between endogenous and exogenous compounds with less extensive decontamination procedures. This is because this approach makes it possible to map the spatial distribution of analytes and examine distribution patterns. By studying these patterns, it can be determined whether a compound is likely to be of exogenous origin (mainly located in the outer structures of the hair) or to have been incorporated via the blood stream (mainly located in the inner structures of the hair). Additionally, by studying distribution patterns along longitudinal sections of hair, temporal records of an individual’s contact with a given substance can be reconstructed. The internal structure of the hair can be exposed via cross- or longitudinal sectioning, and the appropriate choice of sectioning method depends on the purpose of the analysis and the type of sample. 

With this review, we aim to call attention to the unique possibilities offered by MSI and to make the methods more accessible to researchers outside of forensic toxicology. As sample preparation is a critical part of these techniques, we outline how previous studies have addressed the issues related to this step and highlight potential pitfalls. The main point is that sample preparation protocols should be developed and thoroughly evaluated on a case-by-case basis. However, we believe that once appropriate sample preparation protocols have been developed, MSI analysis of hair strands could also find relevant applications within, for example, the biological and medical fields. 

## Figures and Tables

**Figure 1 molecules-26-07522-f001:**
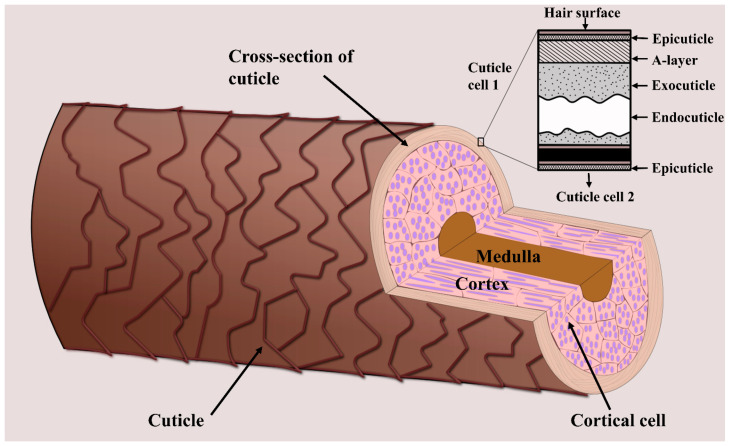
Schematic showing the three main layers of a hair strand—the cuticle, cortex, and medulla. The inner hair structure (i.e., the cortex and medulla) is surrounded by several layers of cuticle cells. The figure was created based on [18].

**Figure 2 molecules-26-07522-f002:**
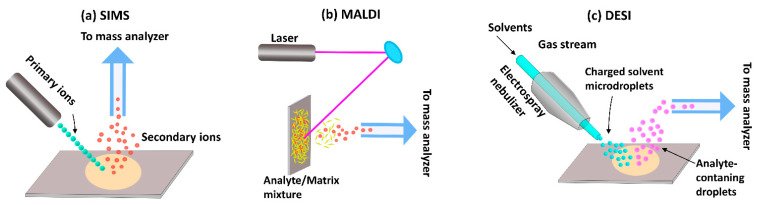
Ionization approaches used in MSI. (**a**) Secondary ion mass spectrometry (SIMS), (**b**) Matrix-assisted laser desorption/ionization mass spectrometry (MALDI-MS), (**c**) Desorption electrospray ionization mass spectrometry (DESI-MS).

**Figure 3 molecules-26-07522-f003:**
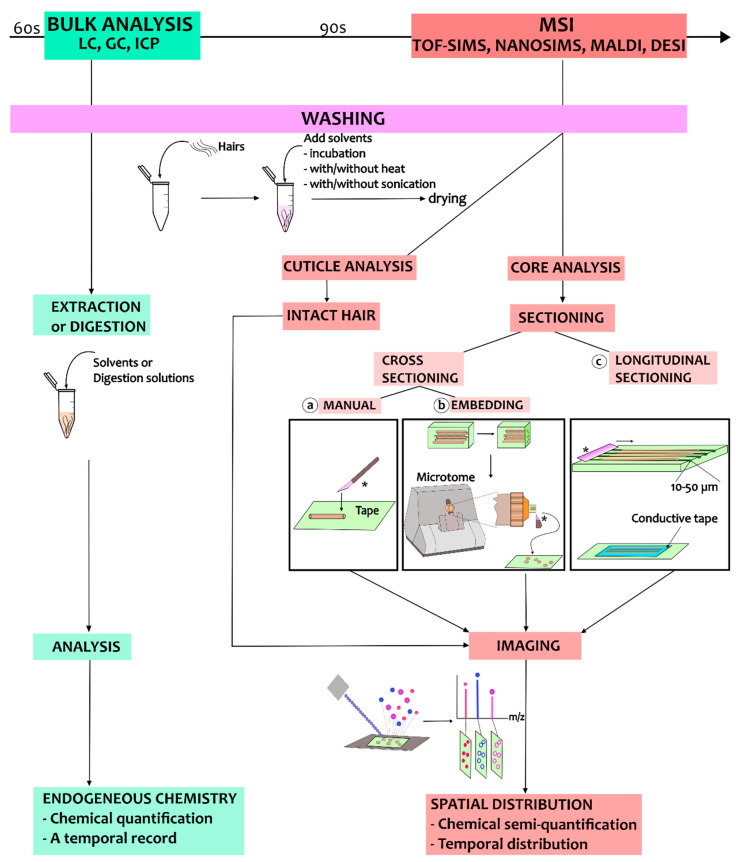
A work chart illustrating the sample preparation process for analysis of hair using MSI. The process starts with the washing steps, and these are followed by different sample preparation procedures depending on the purpose of the MSI analysis. The blade for sectioning is marked by ‘*’.

**Figure 4 molecules-26-07522-f004:**
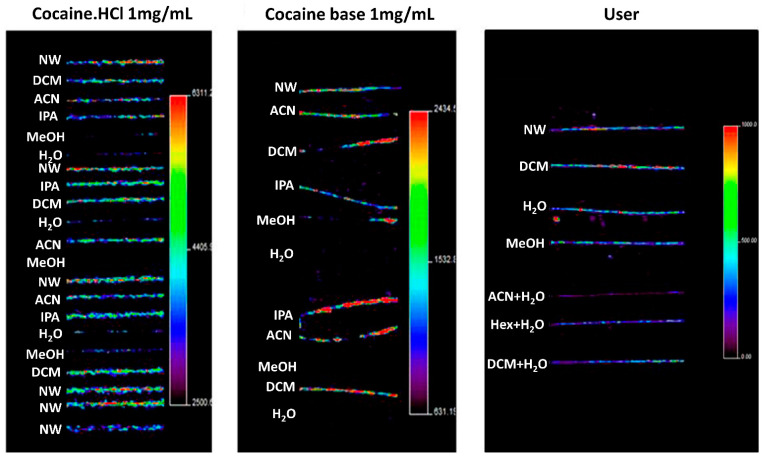
MSI imaging of the cocaine precursor ion in contaminated and users’ hairs, illustrating the effects of washing for 1 min in different washing solutions. Hair strands contained cocaine after being contaminated using a cocaine·HCl stock solution (left panel), using cocaine base (middle panel), and after cocaine use (right panel). The authors noted that washing with water and methanol significantly decreased cocaine concentrations in contaminated samples. However, it was also found that this washing procedure seems to cause migration of external contaminants into the hair cortex and medulla. NW, no wash; DCM, dichloromethane; ACN, acetonitrile; IPA, isopropanol; MeOH, methanol; H_2_O, water; Hex, hexane. Images show a significant reduction in cocaine contamination after washing with methanol, water or both. Figure reprinted and modified from [87]. Copyright (2021) American Chemical Society.

**Figure 5 molecules-26-07522-f005:**
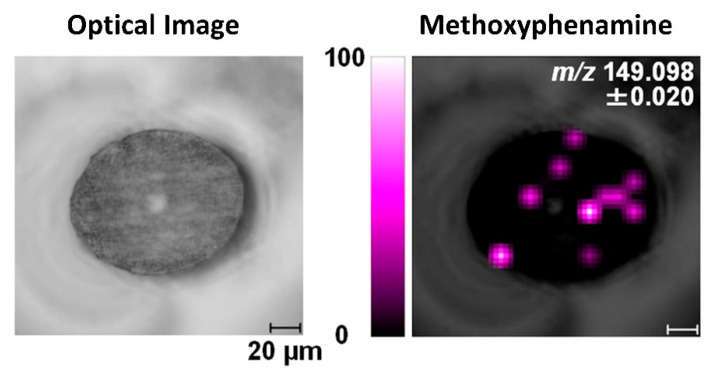
Image of methoxyphenamine in a cross-sectioned hair strand analyzed by MALDI-ion trap TOF MS/MS. The hair is embedded in gelatin, and the spatial resolution is 10 µm. Reprinted with permission from [50]. Copyright (2021) American Chemical Society.

**Figure 6 molecules-26-07522-f006:**
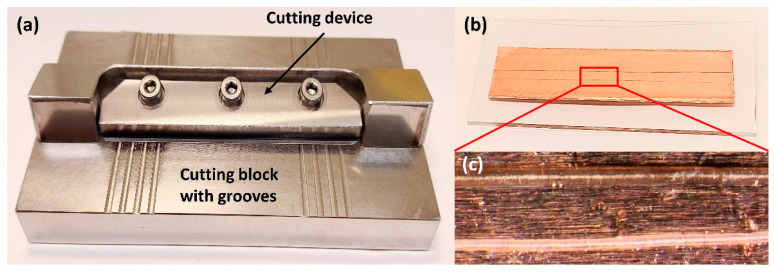
Images show the preparation of longitudinal sections of single hair samples. (**a**) The device with 20–80 µm grooves used for longitudinal sectioning of hair samples; (**b**) Longitudinal hair sections are attached into double-sided copper tape; (**c**) Optical images of the longitudinal sections of single hairs. This device was designed based on references [76,106]. Photos by the authors.

**Figure 7 molecules-26-07522-f007:**
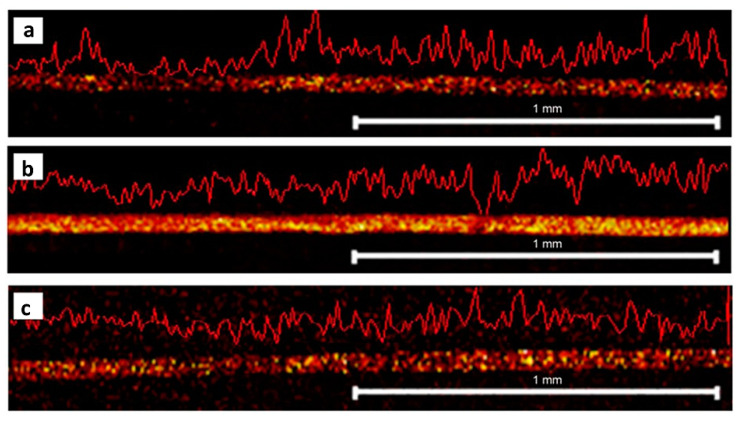
Imaging of drugs in longitudinal hair sections. Metal-assisted SIMS reveals the distribution of (**a**) benzoylecgonine at *m/z* 290, (**b**) cocaine at *m/z* 304, and (**c**) methadone *m/z* 310 in a longitudinal section of a hair strand. The hair was sectioned using a cutting device without embedding. Figure modified and reprinted with permission from [108].

**Figure 8 molecules-26-07522-f008:**
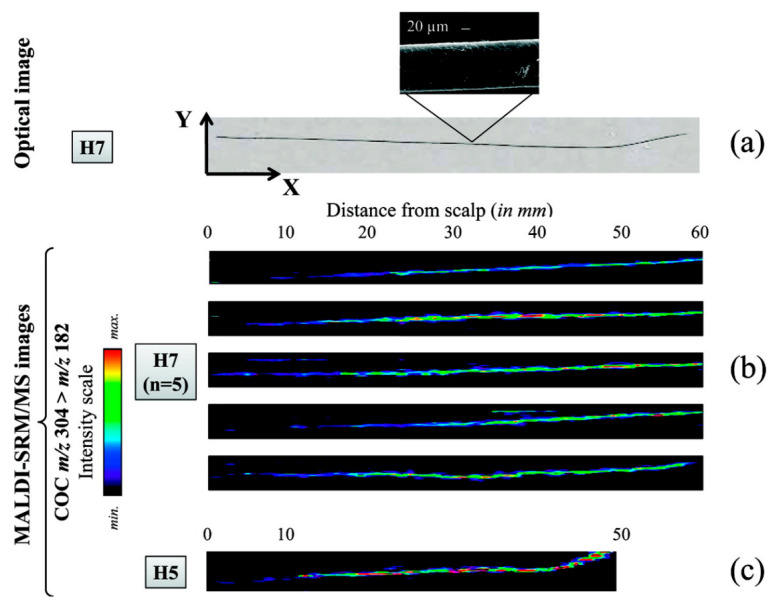
Relative quantification of the record of cocaine consumption over several months, as reflected in single intact hair samples from two different individuals (H7 and H5). The images were obtained by Porta et al. using MALDI-selected reaction monitoring/MS. (**a**) An optical image of a single hair from individual H7, attached to the stainless steel MALDI plate using double-sided adhesive foil. (**b**) Imaging of cocaine distribution in five different intact hair samples from H7, indicating that consumption was probably reduced over the last 6 months (the average growth rate of scalp hair being approximately 1 cm/month [114]). (**c**) An intact hair sample from individual H5, illustrating the lowest cocaine concentration detected (4.9 ng/mg, determined by LC-SRM/MS) within the first 10 mm of sample. Reprinted with permission from [110]. Copyright (2021) American Chemical Society.

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
