# Peer review of "Mapping the Chemistry of Hair Strands by Mass Spectrometry Imaging—A Review"

_molecules, 2021, doi:10.3390/molecules26247522_

Round 1
Reviewer 1 Report
Peer Review: Mapping the Chemistry of Hair Strands by Mass Spectrometry Imaging – a Review
This review covering the various sample preparation methods and exploration of MSI for the analysis of hair was an interesting read. The review remains focussed on the area of hair analysis using MSI for various applications whilst providing enough background of the previously used techniques to show how these complements and justify the use of an MSI approach in this field.
Major Comments
The majority of the review is more of an assessment of the different sample preparation approaches and their subsequent effect on MSI data acquired as opposed to the comparison of MALDI vs. SIMS and the scope for other MSI platforms to be used as you'd maybe think from the title. The imaging methods used for hair analysis are glanced over, with little explanation of how they work or their use. Either change the emphasis of the title and abstract to reflect this or include a broader evaluation of the technology, keeping in mind the intended non-specialist reader.
There is a frustrating lack of specificity in the writing for example the informal tone i.e. line 367 'basically', line 112 'big molecules'. Throughout the document where known specific details like the mass of the molecules and detection limits need to be added in. The non-specialist reader for example will not know the level of detection possible and how that compares to other methods.
Minor Comments
The transferability of this approach to non-forensic specialists in its use is communicated well, but the technical side of the methods is not fully set out.
There is an appropriate use of figures in terms of the sample prep. The two which include data are focused on showing the effects of the different sample prep techniques and not the successful applications of MSI for some of the uses initially detailed (e.g geographical provenance, habitual drug use etc.) the inclusion of such figures would give breath to the work.
1) Fig 1: some greater annotation of this figure would help, for example, the type of molecules found at each location.
2) Fig 3: Some of the chemical formulas do not have the correct superscript (e.g. H2O) (may not be able to change this as the figure is from another publication).
3) Some repetition of words i.e. lines 42, 89, 115, where "for example", "such as, for example" are used but aren't needed.
4) Reference in line 196 to Kempson and Skinner 2012 is not numbered like the other references.
5) 'Sample-preparation' doesn't need to be hyphenated. (Lines 78 and 80)
6) Improved subheadings when the review moves onto MSI discussion and finished the sections on sample preparation e.g. sectioning, embedding. Section 3.2 suggests it is still carrying on from the previous sections but it has actually moved onto MSI in more detail.
7) Possibly include some more figures showing the various successful applications? The current ones only seem to be related to the optimisation/investigation of sample prep techniques. These may be a good addition and help break up the text more.
8) Section 3 should have a subheading underneath as it immediately goes into detail about the challenges with MSI e.g. semi quantitation, effects of matrix, pre-contamination etc. and is not an explanation about MSI as you would think from the heading.
9) Add a description of SIMS and MALDI or a figure showing how these techniques differ for the reader, along with the mass analysers detailed (maybe a table format for easier reading?).
10) Line 142 describes the issue of matrix effect on quantitative experiments- 'thin layer' is quite ambiguous- do you mean a known amount deposited using a sprayer? As depending on the thickness/ type of hair, more matrix may be required depending on this and a thin layer may not be sufficient.
Additional Punctuation
- Line 313: full stop missing.
- Line 347: does not require as opposed to 'do not require'.
- Line 392: 'Hair dying' products should be 'dyeing'.
Papers Omitted:
1) Emma Beasley, Simona Francese, and Tom Bassindale Analytical Chemistry 2016 88 (20), 10328-10334 DOI: 10.1021/acs.analchem.6b03551 - Detection of cannabinoids in hair
2) Wang H, Wang Y. Matrix-assisted laser desorption/ionization mass spectrometric imaging for the rapid segmental analysis of methamphetamine in a single hair using umbelliferone as a matrix. Anal Chim Acta. 2017 Jul 4;975:42-51. doi: 10.1016/j.aca.2017.04.012. Epub 2017 Apr 10. PMID: 28552305. - Detection of methamphetamine in hair
Author Response
Reviewer 1:
This review covering the various sample preparation methods and exploration of MSI for the analysis of hair was an interesting read. The review remains focused on the area of hair analysis using MSI for various applications whilst providing enough background of the previously used techniques to show how these complements and justify the use of an MSI approach in this field.
Major Comments
The majority of the review is more of an assessment of the different sample preparation approaches and their subsequent effect on MSI data acquired as opposed to the comparison of MALDI vs. SIMS and the scope for other MSI platforms to be used as you'd maybe think from the title. The imaging methods used for hair analysis are glanced over, with little explanation of how they work or their use. Either change the emphasis of the title and abstract to reflect this or include a broader evaluation of the technology, keeping in mind the intended non-specialist reader.
Yes, thanks, we agree with the reviewer that this was lacking. Therefore, the principles of different MSI approaches were added in the beginning of section 3.
Traditionally, MSI techniques have been most widely used in the materials sciences. However, since MSI can be used to analyze and visualize any chemical species that can be desorbed and ionized from a sample surface, the methods have since found application within the biomedical and forensic sciences as well. The three major ionization techniques commonly used in MSI are MALDI, SIMS, and desorption electrospray ionization (DESI) (Figure 2).
SIMS was the first ionization technique to be developed and is, therefore, the oldest MSI method [54,55]. In this approach, a high-energy primary ion beam is applied to sputter the sample surface, resulting in the generation of ionized species that are then separated based on their mass-to-charge ratio (m/z) using a TOF or magnetic sector mass analyzer. Traditional SIMS instruments were operated with high doses of a monoatomic ion beam (e.g., Ar+, Ga⁺, and Bi⁺) which damaged the sample surface and produced small fragment species [54,56]. Thus, the detection of intact biomolecules was limited with SIMS-based imaging. However, subsequent evolution of primary ion beams – from monoatomic ion beams to cluster ion beams (Bi3+, Au3+, C60+, and so on) and, later, gas cluster ion beams (Ar4000+ or (CO2)6000+) – has since enabled SIMS analysis of intact molecular ions [57-59]. In cluster ion beams, the kinetic energy is divided between several atoms, resulting in lower energy of individual particles. Consequently, the degree of molecular fragmentation and subsurface damage is reduced, improving the mass range of detection for heavier species up to 2500 Da [60]. However, despite these advantages, SIMS remains of limited utility in the analysis of larger biomolecules, such as proteins and peptides.
While SIMS has gained attraction in recent years, MALDI is, at present, the most commonly used technique for biological applications. For this method, a matrix compound, typically an organic acid, is deposited on the sample surface prior to analysis in order to facilitate desorption and ionization. After a laser beam strikes the matrix-coated sample surface, the matrix molecules absorb the laser energy and convert it into heat energy. A fraction of the matrix molecules from the top layer of the sample surface is then vaporized along with analytes, which are ionized via ion or charge transfer processes [61]. In some cases, ionization efficiency can be improved by so-called derivatization, where the sample chemistry is altered in order to change the properties of the analyte. For example, Beasley et al. demonstrated that in situ derivatization improved ionization efficiency enough to enable the imaging of cannabinoids in single hair samples [62]. The popularity of the MALDI technique reflects its ability to probe a variety of molecules, including lipids, proteins, peptides, nucleotides, and saccharides [63-66]. However, it is to be noted that the application of matrix to the sample surface can interfere with the detection of low-molecular-mass species (<300 Da) [67]. Moreover, reproducibility and spatial resolution is limited by the matrix crystal size, raster step size, and laser beam diameter.
Like MALDI, DESI is an ionization technique suitable for the analysis of biological samples. In DESI, the sample surface is bombarded with electrically charged solvent droplets to desorb analytes of interest, which are ionized using electrospray. The ionized molecules then travel into an inlet capillary towards mass analyzers for analysis. The main advantage of this technique is that desorption and ionization can take place under ambient conditions – in contrast to SIMS and MALDI, which operate under vacuum – without sample preparation or matrix application. Consequently, it is a comparably fast technique that makes it possible to preserve the physical and chemical properties of the sample. Today, DESI has great potential to aid forensic investigations in the screening for and identification of drugs [68]. However, the technique has not been extensively used for MSI analysis of hair, probably because of its relatively low spatial resolution: Recent advances have led to improved resolutions of between 20 and 100 µm [69,70], but the average diameter of adult human hair varies between 45 and 110 µm [18] As the technique evolves towards improved resolution [69], it may find application in future hair analyses.
When planning MSI analysis of hair samples – or any other sample – it is important to choose an appropriate ionization process. As described above, different techniques are suitable for different analytes, with MALDI covering a wider mass range (1–500 kDa) than SIMS. Another consideration is resolution requirements. Because SIMS utilizes an ion beam as opposed to a laser, this method has a spatial resolution as high as 100 nm [71], whereas the highest resolution offered by MALDI is 5 µm [72]. The higher resolution makes SIMS the more suitable technique for imaging the detailed distribution of compounds within the hair structure, but it also increases the risk of obtaining unfocused images that might lead to misinterpretation [17]. When imaging large molecules >2000 Da, MALDI is the best method for hair analysis at present, but the addition of the analyte-specific matrix alters the sample composition and may lead to the relocation of compounds [73]. Because MALDI and SIMS have different strengths and weaknesses, applying both methods makes it possible to obtain complementary information [74]. However, whether the choice falls on one or both of these techniques for a given analysis, it is important to be aware of their limitations and challenges.
There is a frustrating lack of specificity in the writing for example the informal tone i.e. line 367 'basically', line 112 'big molecules'. Throughout the document where known specific details like the mass of the molecules and detection limits need to be added in. The non-specialist reader for example will not know the level of detection possible and how that compares to other methods.
We agree. With regard to unspecific wording, we have rephrased. Also, we agree the difficulty to define detection limits of molecules that are detectable with MSI is unsatisfactory. Here, we have added general explanations as the detection limits in the text. Also, we now emphasize that the in some cases the detection limit is known for e.g., cocaine (see row 500).
Line 367: ‘basically’ was removed (in line 468).
Line 112: a mass range was listed after ‘big molecules’: ‘More recently, MSI has been applied to obtain the spatial distribution of several species on and in hair, ranging from small elements to molecules as large as 4000 Da [41-43]'
Minor Comments
The transferability of this approach to non-forensic specialists in its use is communicated well, but the technical side of the methods is not fully set out.
There is an appropriate use of figures in terms of the sample prep. The two which include data are focused on showing the effects of the different sample prep techniques and not the successful applications of MSI for some of the uses initially detailed (e.g geographical provenance, habitual drug use etc.) the inclusion of such figures would give breath to the work.
1) Fig 1: some greater annotation of this figure would help, for example, the type of molecules found at each location.
Since there is no particular localization pattern associated with specific molecules, we do not think that further annotations should be added into Figure 1.
2) Fig 3: Some of the chemical formulas do not have the correct superscript (e.g. H2O) (may not be able to change this as the figure is from another publication).
Agreed, and we have edited these and clarified that the figure is ‘modified’ from the original. Thanks for pointing this out.
3) Some repetition of words i.e. lines 42, 89, 115, where "for example", "such as, for example" are used but aren't needed.
Indeed, thanks. These words were discarded.
4) Reference in line 196 to Kempson and Skinner 2012 is not numbered like the other references.
Yes, the reference number is now added (now in line 278).
5) 'Sample-preparation' doesn't need to be hyphenated. (Lines 78 and 80)
True, thanks. Hyphenations were deleted.
6) Improved subheadings when the review moves onto MSI discussion and finished the sections on sample preparation e.g. sectioning, embedding. Section 3.2 suggests it is still carrying on from the previous sections but it has actually moved onto MSI in more detail.
Yes, we understand where the confusion is coming from. Thanks. We have now restructured the subheadings, especially after the text addition pertaining to the MSI technique.
7) Possibly include some more figures showing the various successful applications? The current ones only seem to be related to the optimisation/investigation of sample prep techniques. These may be a good addition and help break up the text more.
Yes, this is a good idea and we have added Figures 5 and 7 to visualize different drugs in longitudinal and cross-sections of hair strands.
8) Section 3 should have a subheading underneath as it immediately goes into detail about the challenges with MSI e.g. semi quantitation, effects of matrix, pre-contamination etc. and is not an explanation about MSI as you would think from the heading.
Yes, we have reconstructed the subheading.
9) Add a description of SIMS and MALDI or a figure showing how these techniques differ for the reader, along with the mass analysers detailed (maybe a table format for easier reading?).
See point above on the principles of different MSI approaches were added in the beginning of section 3. We believe that an exhaustive explanation of mass analyzers is beyond the scope of this paper, that aims to introduce the power of MSI to a broader audience.
10) Line 142 describes the issue of matrix effect on quantitative experiments- 'thin layer' is quite ambiguous- do you mean a known amount deposited using a sprayer? As depending on the thickness/ type of hair, more matrix may be required depending on this and a thin layer may not be sufficient.
Yes, thanks. The word ‘thin’ was removed.
Additional Punctuation
Line 313: full stop missing. Unfortunately, we cannot find this missing punctuation.
Line 347: does not require as opposed to 'do not require'. - Corrected
Line 392: 'Hair dying' products should be 'dyeing'. - Corrected
Papers Omitted:
1) Emma Beasley, Simona Francese, and Tom Bassindale Analytical Chemistry 2016 88 (20), 10328-10334 DOI: 10.1021/acs.analchem.6b03551 - Detection of cannabinoids in hair
The reference was added in the text, line 170. ‘For example, Beasley et al. demonstrated that in situ derivatization improved ionization efficiency enough to enable the imaging of cannabinoids in single hair samples [62].’
2) Wang H, Wang Y. Matrix-assisted laser desorption/ionization mass spectrometric imaging for the rapid segmental analysis of methamphetamine in a single hair using umbelliferone as a matrix. Anal Chim Acta. 2017 Jul 4;975:42-51. doi: 10.1016/j.aca.2017.04.012. Epub 2017 Apr 10. PMID: 28552305. - Detection of methamphetamine in hair
The reference was added in line 226. ‘In MALDI, the concept of matrix effects is exploited, and a matrix compound is deposited on the sample surface to mediate desorption and ionization of a given analyte [55]. For example, Wang et al. used umbelliferone to improve the detection limit of methamphetamine in hairs down to nanogram per milligram using MALDI-FTICR [80].’

Reviewer 2 Report
The manuscript "Mapping the Chemistry of Hair Strands by Mass Spectrometry Imaging – a Review" by Philipsen et al. reviews hair samples' preparation protocols used for MSI analyses and highlights the main advantages and drawbacks of discussed methods. It was a pleasure to read the text as it is well written. I think that this manuscript describes well the analysis of hair samples by MSI and will be a good source of information for readers not only from the field of forensic science and analytical chemistry, but also from other relevant fields. It deserves to be published after minor revision/changes pointed out below.
The manuscript can be improved by adding a table where will be summarized different preparation protocols discussed with their main benefits and drawbacks, and also with the information about the analyte and MSI method for which the preparation protocol was used.
Other points to address are:
1) In the legend of Figure 1 can be added a reference(s) from which the authors where inspired in order to create the scheme.
2) In the line 314 the number of subsection is wrong. There should be "3.1.3." and not "3.1.2." – please correct it.
3) In the legend of Figure 4 there is no information about references from which the photos were taken. Are these photos made by authors? Please add there some reference(s).
After these changes/corrections will be made I recommend to accept this manuscript.
Author Response
We thank the reviewers for excellent comments. We have revised the paper according to their suggestions, which have improved the paper significantly. We reply to the reviewer’s comments in detail below.
Reviewer 2:
Imaging – a Review" by Philipsen et al. reviews hair samples' preparation protocols used for MSI analyses and highlights the main advantages and drawbacks of discussed methods. It was a pleasure to read the text as it is well written. I think that this manuscript describes well the analysis of hair samples by MSI and will be a good source of information for readers not only from the field of forensic science and analytical chemistry, but also from other relevant fields. It deserves to be published after minor revision/changes pointed out below.
The manuscript can be improved by adding a table where will be summarized different preparation protocols discussed with their main benefits and drawbacks, and also with the information about the analyte and MSI method for which the preparation protocol was used.
We thank the reviewer for this idea that would be very helpful, if clear. Since preparation protocols are not specific to individual analytical techniques or samples, however, we feel this table would be hard to structure in a clarifying way. Instead, we hope that the revised manuscript more clearly emphasizes that the sample preparation methods must be developed and evaluated for each experiment.
Other points to address are:
- In the legend of Figure 1 can be added a reference(s) from which the authors were inspired in order to create the scheme.
Indeed, thanks. The reference was added in the caption of Figure 1.
Reference:
Robbins, C. R., Chemical and physical behavior of human hair. New York: Springer 2012, 5th ed.
2) In the line 314 the number of subsection is wrong. There should be "3.1.3." and not "3.1.2." – please correct it.
We have restructured the subsections and believe they are correct. Thanks.
3) In the legend of Figure 4 there is no information about references from which the photos were taken. Are these photos made by authors? Please add there some reference(s).
Yes, Figure 4 was made by authors. We also added references in the figure 4.
Reference:
1.Flinders, B.; Cuypers, E.; Porta, T.; Varesio, E.; Hopfgartner, G.; Heeren, R. M. A., Mass Spectrometry Imaging of Drugs of Abuse in Hair. Methods Mol Biol 2017, 1618, 137-147.
2.Kempson, I. M.; Skinner, W. M.; Kirkbride, P. K., A method for the longitudinal sectioning of single hair samples. J Forensic Sci 2002, 47, (4), 889-892.
Round 2
Reviewer 1 Report
The additions to the paper have addressed my comments. I not the specfic laungage used in the new addtions which help to lift the artical. The additional images and figures give the review a more rounded feel.